# An 8000 years old genome reveals the Neolithic origin of the zoonosis *Brucella melitensis*

Louis L'Hôte[1], Ian Light [2], Valeria Mattiangeli[1], Matthew D. Teasdale [1,3], Áine Halpin [1], Lionel Gourichon [4], Felix M. Key [2] & Kevin G. Daly [1,5]

*Brucella melitensis* is a major livestock bacterial pathogen and zoonosis, causing disease and infection-related abortions in small ruminants and humans. A considerable burden to animal-based economies today, the presence of *Brucella* in Neolithic pastoral communities has been hypothesised but we lack direct genomic evidence thus far. We report a 3.45X *B. melitensis* genome preserved in an ~8000 year old sheep specimen from Menteşe Höyük, Northwest Türkiye, demonstrating that the pathogen had evolved and was circulating in Neolithic livestock. The genome is basal with respect to all known *B. melitensis* and allows the calibration of the *B. melitensis* speciation time from the primarily cattle-infecting *B. abortus* to approximately 9800 years Before Present (BP), coinciding with a period of consolidation and dispersal of livestock economies. We use the basal genome to timestamp evolutionary events in *B. melitensis*, including pseudogenization events linked to erythritol response, the supposed determinant of the pathogen's placental tropism in goats and sheep. Our data suggest that the development of herd management and multi-species livestock economies in the 11th–9th millennium BP drove speciation and host adaptation of this zoonotic pathogen.

Brucellosis is the most prevalent bacterial zoonosis globally[1] infecting 1.6–2.1 million people annually, and is caused by members of the non-motile *Brucella* genus. *Brucella* species are typically characterised by their mammalian host range and zoonotic potential, with four species considered to infect humans[2]. Of these, *Brucella melitensis* is the most frequent cause of brucellosis in humans and has a host reservoir of goats and sheep. The remaining three species linked to zoonotic infection are also domestic animal-associated: *B. abortus*, closely related to *B. melitensis* and primarily infecting cattle; *B. suis*, infecting pigs; and *B. canis*, dogs. These and other *Brucella* species comprise a "core" clade of high genomic similarity, also known as "classic" *Brucella* species, distinguished from the genetically distinct non-core or "atypical" *Brucella* which are rarely associated with zoonotic infection[2]. Brucellosis has higher incidence among cohorts frequently in contact with livestock such as pastoralist communities[3], and in regions most dependent on small ruminant husbandry[1].

Considered intracellular pathogens, *Brucella* species evade and suppress the animal host immune system to establish replicative niches within cells such as macrophages[4]. Following systemic distribution, *Brucella* show tropism towards the mammary gland, reproductive organs and, in females, the placental trophoblasts. Placental invasion is followed by massive bacterial replication[5] and disruption of the placenta in pregnant animals, leading to spontaneous abortion. The placenta, foetal remains, and vaginal discharges are vectors of transmission, as is unpasteurised milk of infected animals[6]. Additional symptoms of *B. melitensis* or *B. abortus* livestock infection include weak offspring and low milk yield, contributing to a substantial economic and animal welfare burden. Brucellosis in humans is typically

[1]Smurfit Institute of Genetics, Trinity College Dublin, Dublin 2, Ireland. [2]Max Planck Institute for Infection Biology, Evolutionary Pathogenomics, 10117 Berlin, Germany. [3]Bioinformatics Support Unit, Faculty of Medical Sciences, Newcastle University, Newcastle upon Tyne, UK. [4]Université Côte d'Azur, CNRS, CEPAM, Nice, France. [5]School of Agriculture and Food Science, University College Dublin, Dublin 4, Ireland. e-mail: kevin.daly@ucd.ie

due to *B. melitensis* infection and has low mortality, causing a range of symptoms including undulating fever, general malaise, and osteoarthritis[7]. However, infection in pregnant women is associated with 18.6–73.3% risk of miscarriage and an increased risk of preterm birth[8].

The intensification of sheep and goat management in 11th–10th millennium Before Present (BP) Southwest Asia, has been hypothesised as a fulcrum for *Brucella* zoonotic and epizootic transmission[9] (the latter the spread of disease from animal to animal). High incidence of caprine foetal and perinatal remains are found in Neolithic settlements across Southwest Asia including at Ganj Dareh in the Iranian Zagros[10], where evidence for goat management in the 11th millennium BP is robust[11]. As infected females represent the main transmission source[6], selective culling of young male goats practised in early herding communities could have created conditions permissive for sustained *Brucella* enzooticity[9], potentiating human-spillover events. Skeletal remains of these early herders are suggestive of *Brucella* exposure: they feature a high incidence (>80%) of cranial porotic hyperostosis and a single case of vertebrate lytic lesions[12], potential consequences of chronic brucellosis. Skeletal evidence of human and equine brucellosis (see examples[13,14]) is better attested in more recent contexts, with several cases of possible genetic confirmation (see refs. 15,16).

Genomic recovery of ancient *Brucella* species is inherently challenging, with just two genomes from the mediaeval period reported[15,16]. *Brucella* can persist in the environment for ~1.5 months[17] but lack features which facilitate macromolecular preservation such as capsules, thick cell walls or endospores. Unlike their soil and water-living *Ochrobactrum* relatives, *Brucella* appear to require an intracellular environment to replicate[18]. The localization of *Brucella* during infection is also not amenable to ancient DNA survival due to its greater localization in the reproductive organs, mammary glands, spleen, liver, and bone marrow[19], tissues rarely preserved for palaeogenomic study. Finally the core *Brucella* species are highly similar at a genomic level with an average nucleotide identity (ANI) of >99.5%, in contrast to greater divergence from atypical non-core species (ANI with *B. melitensis*: ~97.7%) and the soil-ubiquitous sister genus *Ochrobactrum* (~81% ANI)[20]. Together, these complicate the retrieval and validation of ancient, intracellular *Brucella* genomes.

Despite these challenges, we report and validate a 3.45X *B. melitensis* genome recovered from a sheep specimen dating to ~8000 BP, from Menteşe Höyük in Northwest Türkiye, the oldest reported livestock pathogen and zoonotic genome to date.

## Results

### Detection and authentication of Neolithic *Brucella*

We screened for core *Brucella* species published ancient ruminant genomics datasets (representing 146 specimens, see Methods) and six newly-reported ovicaprine petrosal specimens (Supplementary Data 1) from Menteşe Höyük (40.278013°N, 29.523417°E; hereafter referred to as Menteşe) using a *k*-mer matching approach[21] correcting for read count and coverage[22]. A single female sheep specimen from Menteşe (Supplementary Figs. 1-3) produced a signal for *B. melitensis* (Fig. 1a), with 71 assigned reads representing a diversity of unique genome regions (*E* value = 0.00023; Supplementary Data 2). We generated two radiocarbon age estimates for the sample (Supplementary Data 3), and used Oxcal4.4[23] to produce a combined calibrated age of 8007−7863 cal BP (2σ range) (Supplementary Fig. 4), securely within the upper layer of Menteşe's Neolithic inhabitation.

We performed additional shotgun sequencing for Mentese6 from libraries treated with Uracil-DNA Glycosylase (UDG) to remove post-mortem damage, and also libraries lacking UDG treatment, thus preserving native damage patterns (Supplementary Data 4). We then assessed the authenticity of the *Brucella* reads using HOPS[24], an ancient DNA pipeline which utilises the alignment-based metagenomic

assigner MALT[25]. Using a subset of sequencing data with sheep-aligning reads removed (9,739,417 reads; see Methods), MALT assigned 20,040 UDG-treated unique reads to the *Brucella* node, with a *B. melitensis* reference showing the highest alignment rate (Supplementary Fig. 5). Over 86% of aligning reads showed zero mismatches (Fig. 1b), suggesting correct assignment to the *Brucella* node rather than spurious alignments from a related genus. A similar high alignment accuracy is evident when restricted to reads with persisting post-mortem molecular damage (Supplementary Fig. 5), supporting an authentic ancient origin of the *Brucella*-assigned reads. Together, these suggest the Mentese6 genome represents a Neolithic *B. melitensis*.

Aligning UDG-repaired reads to the *B. melitensis* reference genome (GCF_000007125.1) produced a 3.45X alignment (Table 1) after removal of sheep-aligned reads and other quality controls (see Methods). Approximately 94% of the *B. melitensis* genome was covered by at least one read (the 1X genome breadth; Fig. 1c), further supporting this as a genuine *B. melitensis* genome rather than a false positive signal due to alignment from related environmental species. Such spurious alignment can result in "read stacking" at homologous genome regions, leading to poor genome breadth relative to the average depth of coverage across the reference. Nevertheless, we observed some limited regions of stacking in the Mentese6 alignment, which are filtered from downstream analyses through a per-site read maximum and sliding window coverage filter (see Methods). Lastly, aligning reads were short (Fig. 1e, mean 54 bp) and showed terminal transitions damage in both UDG repaired and non-repaired libraries (Fig. 1d), consistent with authentic ancient *Brucella* DNA recovery.

### Mentese6 *Brucella* genome is basal to all known *Brucella melitensis*

We evaluated the phylogenetic position of the Mentese6 *Brucella* genome using several complementary approaches. A maximum likelihood (ML) phylogeny of *B. melitensis*, representing the clades observed today[26] (Supplementary Data 5), and the cattle-associated sister species *B. abortus*, aligned to and rooted on the outgroup *B. suis* places the Mentese6 *Brucella* as a basal member of the *B. melitensis* clade (Fig. 2a). The genome falls close to the ancestral node of *B. melitensis* and *B. abortus*, suggesting it existed relatively soon after separation of the two lineages and their speciation.

The genomes of *B. melitensis* and *B. abortus* are highly similar (pairwise >99% ANI[20]), and there remains a potential effect of soil-living *Ochrobactrum* reads in the Mentese6 alignment. To assess this and confirm the *B. melitensis* species assignment we determined variant sites which discriminate specific lineages or branches within the *Brucella* genus (Supplementary Data 6), where all other genus members carry the alternative allele (Supplementary Fig. 6), using data aligned to *B. melitensis*. Across these clade-defining sites and applying strict filtering (see Methods), the Mentese6 base matched 99.7% of variants fixed in and common to *B. melitensis* and *B. abortus*, and 28.1% variants defining the *B. melitensis* branch (Supplementary Fig. 7, Supplementary Data 7). In comparison, Mentese6 matched 0% variants specific to *B. abortus*, 0% variants diagnostic of *B. canis*, and 0.001% of *B. inopinata* variants. We obtain similar results when aligning to a reference genome of a species (*B. suis*) equally related to *B. melitensis* and *B. abortus*, or using random read sampling rather than a consensus approach to assess possible heteroplasmy (Supplementary Data 7). Finally, ANI values[27] (Fig. 2b, Supplementary Data 8) indicate that the Mentese6 genome has greater mean similarity with *B. melitensis* than *B. abortus* (mean 99.870 vs 99.856; Wilcox rank sum test *p*-value < $2.2 \times 10^{-16}$). We therefore conclude that the Mentese6 genome is a basal representative of *B. melitensis* existing soon after the species split from *B. abortus*, before it accrued many of the mutations fixed in *B. melitensis* today.

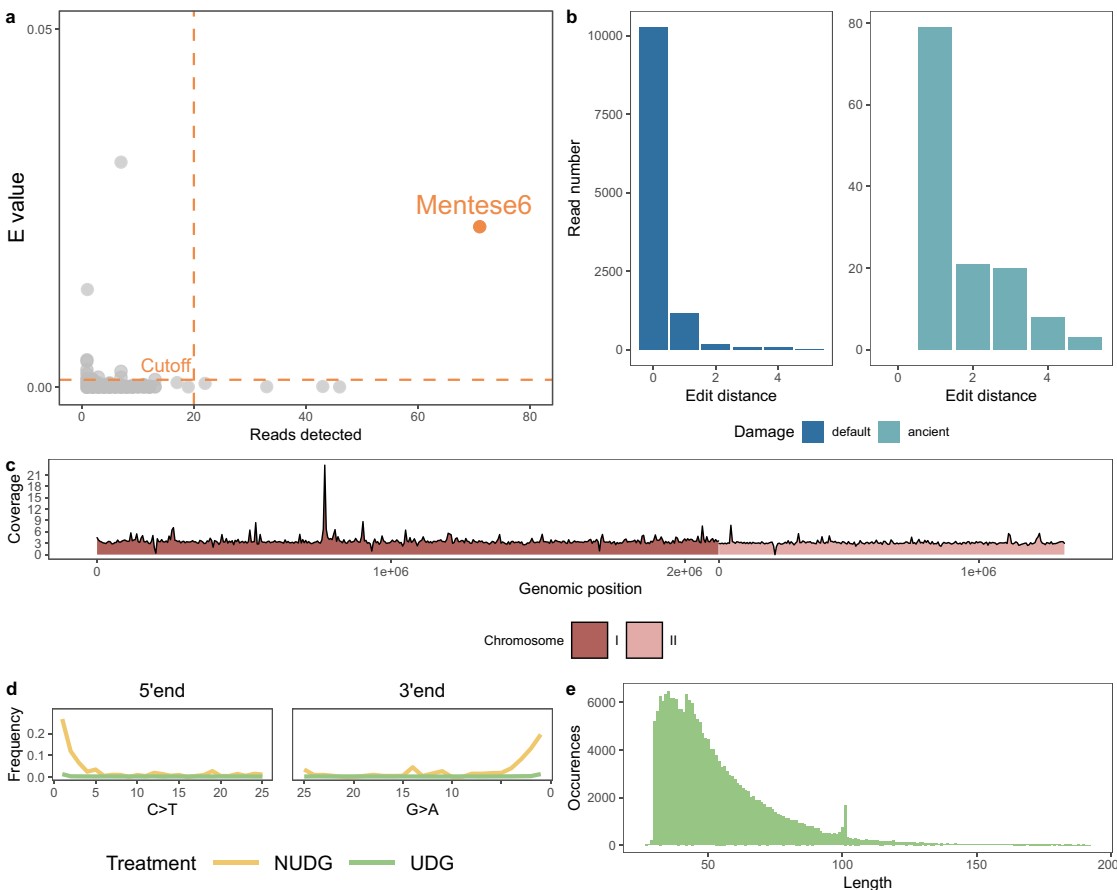

**Fig. 1 | Authentication of ancient *Brucella* DNA in Neolithic sheep specimen Mentese6. a** *k*mer-based *E* value[22] relative to the number of reads detected for core *Brucella* species by KrakenUniq. Highlighted in orange is the screened library (from Mentese6) passing the threshold of > 20 reads and an *E* value > 0.001, indicated by the cutoff dashed lines. **b** HOPS edit distance for Mentese6 Uracil DNA-glycosylase (UDG) treated for reads assigned to the *Brucella* node, merged from seven single-end libraries. Dark blue represents all reads ("default") and light blue reads showing post-mortem damage ("ancient"). **c** Non-overlapping sliding windows (5 kb) of coverage of Mentese6 UDG libraries aligned on *B. melitensis*. **d** mapDamage[68] damage patterns for Mentese6 reads aligned to *B. melitensis*. Green represents UDG libraries and yellow represents non-UDG libraries (NUDG). **e** Read length distribution of the Mentese6 UDG libraries aligned on *B. melitensis*.

**Table 1 | Summary of Mentese6 alignment to *Brucella melitensis* genome from UDG-treated sequencing libraries**

| Sample ID | Reads aligned | Alignment rate (%) | Coverage | 1X Breadth | 3X Breadth | Combined calibrated age (2σ) |
|---|---|---|---|---|---|---|
| Mentese6 | 212,160 | 0.009 | 3.453X | 0.939 | 0.600 | 8007–7863 cal BP |

Values are after removal of duplicate reads and mapping quality 30 filter. 1X and 3X breadth are defined as the proportion of the genome covered by at least 1 or 3 reads respectively.

## Calibrating the *Brucella melitensis* speciation time

We leverage this ~8000 year old genome to refine the speciation time of *B. melitensis* and the primarily cattle-infecting *B. abortus*. Time-calibrated phylogenetic trees are premised on the consistent accrual of mutations through time, creating a relationship between genome sampling time and genetic change: this permits the estimation of timing of individual nodes within a phylogenetic tree. Similar to a previous study[16], we find weak evidence for such a temporal signal[28] in both species (Supplementary Fig. 8). This discrepancy in sampling age of modern *Brucella* genomes and their measured genetic differentiation appears to be driven by clade-specific differences in mutation rate, indicated by clustering of lineages by their root-to-tip distance values. This is further illustrated by inflated ANI values of the *B. melitensis* Western Mediterranean lineage with the Mentese6 genome (Supplementary Fig. 8), leading to Mentese6 being relatively closer to some *B. abortus* sequences (Fig. 2b). Despite this apparent rate heterogeneity, marginal likelihoods supported a strict clock mutation model over a relaxed model which would allow rate to vary

among branches (Supplementary Data 9). We proceeded with Bayesian time tree computation[29] using a data set of *B. melitensis* and a single *B. abortus* as an outgroup, and achieved good model stabilization (all parameter Effective Sample Sizes ≥ 1260; Supplementary Data 9). Our estimate of the *B. melitensis-B. abortus* speciation time is 9816 BP (95% Highest Posterior Density (HPD): 10,196–9447 BP; Fig. 3a), a ~750 year interval within the Neolithic era of Southwest Asia. This period overlaps with the emergence of herding[30], and subsequent consolidation and dispersal of multi-species livestock economies in the region following more than a millennia of animal management experimentation[31].

We infer a common ancestor time of *B. melitensis* today of 4317 BP (4496–4137 BP), within the Bronze Age of Western Eurasia, relatively recent compared to the beginnings of livestock domestication in the 11th millennium BP. A potential explanation is demographic processes in either of its main hosts, sheep and goats: *Brucella* is an intracellular, monoclonal genus, with a closed genome showing limited evidence of recombination[32] susceptible to host-mediated bottlenecks[33].

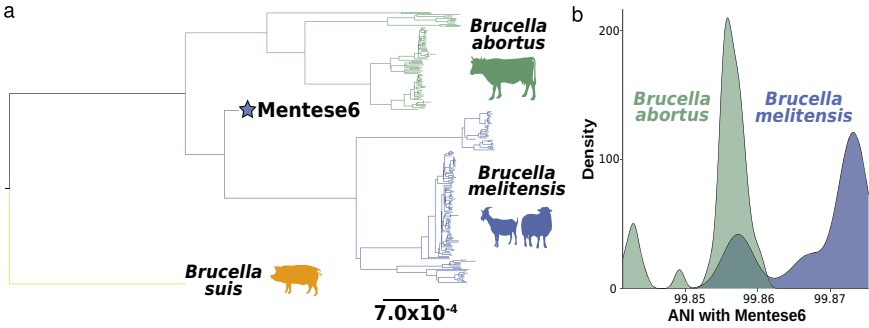

**Fig. 2 | Genome based assignment of Mentese6 to *Brucella melitensis*. a** ML phylogeny (IQ-TREE2[67]) for *B. melitensis* and *B. abortus* aligned to outgroup *B. suis*. The blue star indicates Mentese6. The typical animal hosts of each lineage are shown in silhouette. **b** Distributions of Average Nucleotide Identity (ANI) between Mentese6 and *B. melitensis* (*n* = 144) or *B. abortus* (*n* = 83). *B. melitensis* genomes with lower ANI with Mentese6 (<99.86%) derive from the Western Mediterranean clade, which shows elevated root-to-tip distances (Supplementary Fig. 8) indicative of an elevated mutation rate within that lineage.

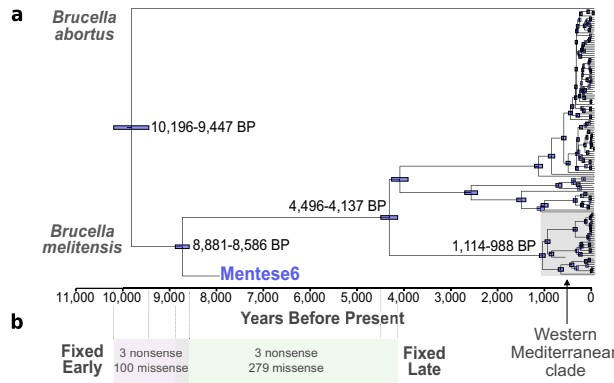

**Fig. 3 | Time-calibrated phylogeny and gene evolution of *Brucella melitensis*. a** Beast[29] phylogeny using a strict molecular clock and Coalescent Bayesian Skyline tree prior. The 95% Highest Posterior Distribution (HPD) are indicated blue, and annotated if mentioned in the text. The *B. abortus* reference genome was included to estimate the *B. melitensis*-*B. abortus* speciation time. Samples were aligned to the *B. melitensis* reference genome. x-axis is Years Before Present (BP - relative to the year 1950). **b** Missense and nonsense mutations fixed in *B. melitensis* today are either early evolving or late evolving, occurring before or after the divergence of the Mentese6 Neolithic genome from the ancestor of modern *B. melitensis*. Dashed lines indicate the extent of the early or late evolving epochs, based on the 95% HPD of node ages from the Beast phylogeny.

This dating predates the occurrence of sheep wool use in Europe[34] and thus could be driven by demographic processes in the host ruminants, although other explanations are possible. Selection may have acted on beneficial mutations and lead to clonal expansion, a selective sweep[35], explaining the relatively recent common ancestor of modern *B. melitensis*. We additionally infer a narrow diversification time (1114–988 BP; 836–962 Common Era) of the Western Mediterranean clade, basal to other modern *B. melitensis* lineages[16]. Our mutation rate estimate of $8.418 \times 10^{-8}$ (95% HPD: $8.770 \times 10^{-8}$–$8.074 \times 10^{-8}$) is a more precise refinement of previous estimates for *B. melitensis*[16].

## Time-stamping functional evolution of *Brucella melitensis*

The ~8000 year old *B. melitensis* genome offers the opportunity to directly time-stamp key evolutionary steps of this zoonotic pathogen. We annotated single nucleotide variants fixed in modern *B. melitensis* relative to other *Brucella*, and identified 6 positions with nonsense mutations covered by at least one read in the Mentese6 Neolithic *Brucella* (Fig. 3b, Supplementary Data 10). At 3 of these 6 nonsense mutation sites, Mentese6 already carried the derived allele, while the remaining 3 lacked the stop codon-introducing allele. This establishes three pseudogenization events (evolutionary changes often associated

with host specificity[32]) before the divergence of Mentese6 from the ancestor of *B. melitensis* today ~8731 BP (95% HPD: 8881–8586 BP) but after the *B. melitensis*-*B. abortus* split (10,196–9447 BP).

The three pseudogenization events occurring prior to the split of Mentese6 from other *B. melitensis* have well defined molecular roles: *nosZ*, *iolG_6* and *hipO*. Of these, *nosZ* is associated with the microbe's mode of living and pathogenesis. *nosZ* encodes nitrous-oxide reductase and is the final step in the denitrification cascade, a respiratory pathway employed under anaerobic conditions. *B. melitensis* is particularly limited in its capacity for anoxic growth compared to other *Brucella* species, a suspected consequence of the *nosZ* pseudogenization and resulting build-up of nitrous oxide and other toxic by-products of anoxic nitrogen reduction[36]. The intracellular environment and compartments have a lower oxygen concentration than the extracellular space[37]. Despite *B. melitensis*' reduced capacity for respiration within the cell, it, along with other core *Brucella* species, is traditionally considered to be a facultative intracellular parasite. *B. melitensis* preferentially replicates extracellularly in the presence of erythritol, a 4 C sugar to which the bacteria shows tropism[38]. Erythritol is produced by ruminant placental trophoblasts, and has been recently shown to be available in human trophoblast culture[39], a potential mechanism of its placental colonisation in humans. The reduced ability of *B. melitensis* to grow under anaerobic conditions, fixed early in its evolution by the *nosZ* pseudogenization, may contribute to its pathogenesis by weighting replication towards the extracellular aerobic spaces when induced by placental erythritol.

For the remaining three pseudogenes (Supplementary Data 10) we observed the ancestral allele in the Mentese6 *Brucella* genome (i.e. the gene lacks the stop codon mutation), and thus postdate its divergence from the ancestor of *B. melitensis* today (95% HPD: 8881–8586 BP). Among these later-fixing nonsense mutations is one affecting BME_RS15065 (previously BMEII1002), truncating this hemerythrin domain-containing protein by 20 residues (137aa compared to 157aa in *B. abortus 2308*). The gene is predicted to have calcium/iron binding activity and is upregulated during growth in response to erythritol[38]; this pseudogenization may have become fixed in *B. melitensis* as a consequence of its erythritol tropism and induced extracellular replication. The remaining two nonsense mutations affect a gene encoding a hypothetical peptidoglycan-binding domain protein (BME_RS05080) and a known glycosyltransferase pseudogene (BME_RS17595). We additionally assess the Mentese6 genome's complement of infection-related genes based on gene-level coverage and breadth relative to the genome average (Supplementary Data 11). These suggest that the Neolithic *B. melitensis* had a sufficient repertoire of genes to establish chronic or congenital infection in its sheep host, enabling genome recovery from the Mentese6 temporal bone.

We similarly identified 379 fixed protein-altering missense mutations, 100 of which were present in the Neolithic Mentese6 *Brucella* genome (Supplementary Data 10) and thus occurring early in *B. melitensis* evolution. Among these are three affecting infection-related genes *fliF*, *lptA*, and *cysB*. *FliF* is a flagellum-related pseudogene in *B. melitensis* today, but we lack reads overlapping the diagnostic nonsense mutation (II:164,232) to establish if this ring monomer protein gene was functional in Mentes6. *LptA* is involved in the modification of the "lipid A" cell envelope component, but this may not affect virulence[40]. Finally, *cysP* (BMEI0673) produces a sulfate/thisulfate-binding protein and is regulated by the key infection-related operon *virB*[41]. We additionally find missense mutations in four erythritol (*ery*) operon genes, fixed in the Neolithic genome and *B. melitensis* today (Supplementary Data 10). Contrasting these are 279 missense mutations fixed later in *B. melitensis* evolution, affecting key genes involved in infection (*virB4*, *virB6*, *virB11*, *virB12*), intracellular defence evasion (*sodB*) and survival in the stomach's low pH environment (*ureC, ureD_2,* and *ureF_2*).

## Discussion

We report an approximately 8000 year old *B. melitensis* genome that confirms previous genomic[16], palaeopathological[12], and transmission model inferences[9] that livestock-infecting *Brucella* were present and sustained in Southwest Asian managed herds already during the Neolithic period. The Neolithic intensification of the human-animal relationship and shift towards pastoralism is considered the first epidemiological transition due to the increased opportunities for zoonotic transmission compared to hunter-gatherer societies[42]. This is supported by human-derived palaeomicrobial evidence[43], although a broad ancient pathogen DNA screen suggests zoonosis frequency only accelerated during the last 7000 years[44]. Here we present direct evidence that livestock host-specific zoonotic pathogens evolved along with the development of livestock herding and multi-species animal-based economies 10,000-9000 years ago (Fig. 3a).

Menteşe is a Pottery Neolithic site located in the eastern Marmara region and was inhabited during the second half of the 9th millennium BP by farming communities primarily engaged in raising sheep and cattle, and to a lesser extent herding goats and pigs[45]. They further supplemented their diets through hunting wild mammals and birds, and gathering terrestrial and freshwater molluscs[45]. After an early focus on cattle, the proportion of sheep at Menteşe rose substantially, indicating small ruminants had a greater role in the food economy at the time when the Mentese6 sheep lived. Mortality profiles, commonly used in zooarchaeology to infer livestock management strategies, indicate that sheep and cattle were exploited not only for meat but also for milk production. Dairy consumption at that time is also supported by lipid residue analysis of pottery sherds from neighbouring early Pottery Neolithic sites[46], suggesting an emphasis across northwestern Anatolia on this practice. Sustained enzootic *Brucella* transmission is likely due to the social and economic interactions between western Anatolian communities[47,48], which would have maintained reservoir populations[9]. These farming communities would have been at risk of brucellosis due to herd-circulating *B. melitensis*: human skeletons from Menteşe show a high frequency of osteoarthritis and anemia[49] indicative of infection, possibly brucellosis. While it is possible the *Brucella* genome recovered from this site was not infection-related, the genus is not considered to be free-living[18]. As the European Neolithic farming tradition stems from Anatolia[30], *B. melitensis* likely also affected the first livestock of Europe.

We refine previous *B. melitensis* and *B. abortus* speciation time estimates[16,50] to a narrow ~750 year window (10,196–9447 BP) during the Neolithic era of Southwest Asia. Our inferred speciation time is consistent with recent estimates drawn from *B. melitensis*[16] and *B. abortus*[50], falling in the early 10th millennium BP, a period documenting the consolidation of livestock-based economies in Southwest Asia. In the Iranian Zagros Mountains, our estimate overlaps evidence of established, multi-generation communities of goat herders ~10,000 years BP, with the appearance of domestic-morphology sheep hundreds of years later[11]. In Central Anatolia, sheep and goat management was practised in the 11th millennium BP[51]; the management of cattle herds is documented by the mid-9th millennium BP, and likely earlier[52]. By the early 9th millennium BP, pastoralist communities managing cattle, sheep, goat, and pig herds spread rapidly along the southern and western coasts of Anatolia[47], eventually into Europe.

Given our estimated *Brucella* diversification times, the intensification and spread of livestock husbandry may have also precipitated an epizootic transition: an increase in the potential for pathogen transmission within the herd and host switching between livestock species. Bacterial shedding due to brucellosis-induced abortion may have been particularly consequential in the context of greater host densities of Neolithic herds[53]. Within-settlement penning and confinement[10], suboptimal hygiene practices in incipient herding communities, and the mixing of previously allopatric species may have promoted transmission and pathogen diversification. The common ancestor of *B. melitensis* and *B. abortus* evolved in this context, presumably diverging from a generalist infection capacity to more host-adapted lineages via circulation in mixed ruminant herds.

The specific genes with fixed nonsense in *B. melitensis* may reflect its pathogenesis within animals and humans. The early-evolving *nosZ* nonsense variant may be a consequence of the *B. melitensis*' prolific extracellular replication induced by erythritol, a step in *Brucella*'s transmission via placental and foetal remains. As *B. abortus* and other genus members show erythritol sensitivity[54], the *nosZ* pseudogenization likely occurred in the context of already-evolved erythritol tropism or response. Selection for respiratory activity under anaerobic conditions could therefore be relaxed in *B. melitensis*, given the tenfold bias to extracellular replication in the presence of erythritol[38]. The late-occurring BME_RS15065 pseudogenization may similarly have been fixed in this context. Additional palaeogenomes, of greater completeness and representing a range of temporal periods, would further refine the tempo of these evolutionary events underlying *B. melitensis*' host adaptation and zoonotic capacity.

## Methods

### Sample provenance

Menteşe Höyük is found in the Yenişehir plain south of Lake Iznik (approximately 40.277 N, 29.524E), and is one of the earliest Neolithic sites in northwest Anatolia, occupied from the 9th millennium BP[55]. The Menteşe Höyük specimens analysed here derive from late-9th millennium BP settlement layers, from the 2000 test pits was exported in 2002 to Lionel Gourichon with the permission of the director of the Iznik Museum, Taylan Sevil, and the site's excavation directors, Jacob Roodenberg and Songül Alpaslan-Roodenberg. Consent for the genetic analysis, as part of the ERC CODEX project, was given in August 2013 by Ömer Eren, then director of the Iznik Museum, with the approval of the excavation directors.

### Sampling and extraction

All laboratory work was performed in dedicated ancient DNA facilities in Trinity College Dublin. We sampled five ovicaprine petrosal specimens from Menteşe Höyük (Supplementary Data 1). Petrous bones were decontaminated by 30 minute exposure to UV light and removal of the surface dirt using a drill bit. A dremel saw was then used to subsample the petrous bones at the densest region. Bone subsamples were pulverised using a MixerMill at 30 f, until complete powderization.

100–150 mg of bone powder was used for DNA extraction. Initial screening double-stranded DNA (dsDNA) libraries were constructed from DNA extracted according to silica-based extraction protocols[56]. We performed an additional extraction round on bone pellets

remaining from the original extraction incubation. Bone pellets were subject to an EDTA-proteinase K (Fisher, 10628203; Sigma-Aldrich, 3115828001) digest at 37 °C for 24 hours using a modified pH PB buffer[57].

## Library building and shotgun sequencing

All libraries were dsDNA libraries constructed following the Meyer and Kircher protocol[58]. One library using 5 μl starting DNA was prepared without uracil DNA glycosylase treatment, to capture native damage patterns. All other prepared libraries were pre-treated with 5 μl UDG-glycosylase (New England Biolabs, M5505L) at 37 °C for 1 hour. Following dual indexed-PCR amplification for 12 cycles, libraries were shotgun sequenced on HiSeq 2000 (Macrogen, Seoul), MiSeq and NovaSeq 6000 platforms (TrinSeq, Dublin) (Supplementary Data 4).

## Radiocarbon dating

We generated two radiocarbon age estimates for Mentese: UBA-47124: 7066 + / 42, and OxA-43559: 7117 + /− 25 (Supplementary Data 3). We then used R_Combine and the Oxcal4.4[23] to produce a combined calibrated age distribution. The resulting calibrated age 2σ range was 6057-5913 cal BCE (8007-7863 cal BP).

## Species and sex determination

Screening libraries from Menteşe were assigned species based on Fastq Screen[59]. Mentese6 was assessed to be a sheep based on total assigned reads, and the "one read, one genome" metric (Supplementary Fig. 3). We assessed karyotypic sex based on the ratio of reads aligning to chromosomes and chromosome length (Supplementary Fig. 2).

## Screening using KrakenUniq and E value

After deduplication with PRINSEQ[60], we screened one fastq file per sample from available ancient ruminant datasets (PRJEB26011, PRJEB31621, PRJEB40573, PRJEB43881) and from Menteşe specimens. Following adaptor removal with cutadapt[61] (single end reads) or AdapterRemoval v2[62] (paired end reads), fastq files were cleaned from host reads by aligning libraries with bowtie2[63] on a concatenated fasta mix of different animal and human reference genomes (GRCh38, Sscrofa11.1, ARS-UCD1.2, mCerEla1.1, ASM170441v1, EquCab3.0, ARS-UI_Ramb_v2.0) with no mapping quality filter applied. Then, the metagenomic profile of the non-aligned reads was computed with KrakenUniq[21] (v1.0.4) using a custom database containing a microbial version of NCBI nt combined with human and complete eukaryotic reference genomes[64].

The KrakenUniq algorithm is effective at identifying low abundant taxa reads; notably, KrakenUniq reports unique $k$-mer assigned to reads and an estimate of the breadth of coverage. Guellil and colleagues[22] developed a score to discriminate true and false positives in taxonomic assignment using KrakenUniq based on an E value calculated as: $E = (K/R)*C$, where K is the unique $k$-mer count, R is the read count and C is the estimate of breadth of coverage. The E value of each KrakenUniq metagenomic assignment was calculated and the results were filtered at the species taxonomic level for a list of core Brucella (B. abortus, B. canis, B. inopinata, B. melitensis, B. suis, Brucella microti, B. ovis, B. pinnipedialis, B. neotomae and B. ceti). We further investigated Mentese6, the only library passing our threshold of minimum 21 reads assigned at species level and E value > 0.001 (Supplementary Data 2).

## HOPS

As Mentese6 passed the E value and minimum read number threshold, we assessed the authenticity of the Brucella signal with the HOPS pipeline[24], using seven merged single-end fastq files due to memory limitations. We used as input the default pathogen list (https://github.com/rhuebler/HOPS/blob/external/Resources/default_list.txt), in addition to all core Brucella species listed above and other Brucella genus

members. A reduced RefSeq database was constructed which minimises the inclusion of redundant sequences while maintaining the observed diversity within species. All available assemblies at the chromosome and whole genome level were considered for bacteria, archaea and viruses. For species which had more than one genome, all genomes were subjected to the MeshClust2 software, which utilises the mean-shift algorithm to select representative sequences that encompass distinct sections of the nucleotide diversity[65]. Selected genomes were subjected to a blast search on the first 10,000 bases to check for mislabelled genomes. Blast-matches to the sequence of interest were ignored and the genome was removed if any of the top 3 bitscore hits were not within the same species taxid. Several eukaryotic genomes were included as representation of species likely associated with domestication contexts. Conterminator was run to identify and remove eukaryotic contigs that were very short or significantly (>5%) contaminated across kingdoms[66]. The database was built using a step size of 4 to decrease the necessary computational resources.

## Lineage/diagnostic variant assessment

To assess the species-level identity of the Mentese6 Brucella, we downloaded published genomes[26] and confirmed species-level assignment with a core genome alignment relative to B. melitensis (GCF_000007125.1) using snippy (https://github.com/tseemann/snippy) and a IQTREE-2[67] ML phylogeny (parameters: -T 8 -m MF). We amended species assignment based on phylogeny inspection, and then reduced the dataset to the closely related "classic" or core Brucella species (B. suis, B. canis, B. ovis, B. abortus, B. melitensis, B. ceti, B. pinnipedalis, B. microti, B. neotomae) and a clade containing B. inopinata. For B. suis, B. abortus, and B. melitensis, we randomly subsampled genomes to 23 representatives, matching the number of B. canis genomes (Supplementary Data 6). The resulting 141 genomes were used to generate a core SNP alignment relative to either the B. melitensis or B. suis (GCF_000007505.1) reference genomes, using the software snippy. For both reference-based alignments, a custom python script was used to identify variants which were fixed for a derived allele in a given lineage (e.g. B. canis; the B. melitensis-B. abortus clade), and the ancestral allele was observed in all other Brucella (extract_lineage_sites.py; all custom scripts available at https://github.com/LouisLhote/Neolithic_Brucella_paper/). For these sites, a pileup file was generated for the Mentese6 (samtools mpileup -q 30 -Q 20 -B). A custom python script (pseudohaploidize_pileup.py) was then used to randomly sample a base at each site. In parallel, we applied a more stringent call approach by removing sites with more than one allele observed (i.e. possible heteroplasmy) and with 10 or more covering reads, using a custom python script (pseudohaploidize_pileup_no-heteroplasmy-max10.py). Finally, the observed Mentese6 alleles were compared with the derived and ancestral alleles to calculate a lineage match rate (Supplementary Data 7), using a custom R script (compare_lineage_sites_pseudohap-pileup.r).

## Alignment of Mentese6 data to host and Brucella genomes

Adaptor trimming, and quality filtering of demultiplexed reads was performed using AdapterRemoval v 2.3.2[62], removing reads with <30 bp length and combining overlapping read pairs (−collapse −minadapteroverlap 1 −adapter1 AGATCGGAAGAGCACACGTCTGAACT CCAGTCAC −adapter2 AGATCGGAAGAGCGTCGTGTAGGGAAAGAGT GT −minlength 30 −trimns −trimqualities).

To remove host reads, the collapsed reads were aligned on the sheep reference genome oviAri3 (GCA_000298735.1) using the bwa aln algorithm with relaxed parameters (-l 1024 -n 0.01 -o 2). All non-aligned reads were then aligned on the B. melitensis reference genome ASM712v1 (GCF_000007125.1) with the same parameters as above. Reads with mapping quality under 30 were filtered out with samtools v1.19.2, and duplicates were removed with picard MarkDuplicates v2.26.11. An additional alignment was performed of all non sheep-

aligned reads with the same parameters on the *B. suis* reference genome ASM750v1 (GCF_000007505.1). We calculated post-mortem damage patterns on the *B. melitensis* alignment using mapDamage[68]. For downstream analysis only USER-treated data were used.

## Gene level coverage and breath

Coverage and breadth (the proportion of bases covered by at least one read) was estimated for each gene in the *B. melitensis* bv. 1 str. 16 M strain reference genome (GCF_000007125.1) using the RefSeq annotation and bedtools v2.31.1 coverage (Supplementary Data 11). A 5kbp sliding window of coverage was also computed using bedtools coverage (Fig. 1c). The coverage peak of ~21X on chromosome I overlaps with two RNA polymerase genes, removed in downstream analyses through a pers-ite maximum coverage and sliding window percentile coverage filter. Low coverage regions both overlap ribosomal RNA and tRNA genes.

## Phylogenetic analysis and reference datasets

We first assessed the phylogenetic position of Mentese6 using alignment to *B. suis*, an outgroup to *B. melitensis* and *B. abortus*. We downloaded a reference dataset representing global diversity of *B. melitensis* and *B. abortus* genomes[26] and then removed sequences which lacked a known sampling year and after thinning the dataset with treemmer (-RTL 0.95; available at https://github.com/fmenardo/Treemmer) following the approach of Long and colleagues[15,16], leaving a final set of 227 genomes (Supplementary Data 5). The 227 genomes were converted into fastq reads using ART[69] with a read length of 150 bp and a read coverage of 20 fold. The same parameters for the Mentese6 read alignment step were used to align those reads on the *B. suis* reference genome ASM750v1 (GCF_000007505.1). A vcf file was produced for every genome and for the Mentese6 bam file with GATK package (available at https://gatk.broadinstitute.org/hc/en-us), using the "EMIT_ALL_SITES" and "PLOIDY = 2" option that generated a call for all positions in the reference genome.

As intracellular *Brucella* have many free-living, closely-related relatives, we aimed to minimise false positive variant calling due to spurious alignment and retain only high confidence variants for Mentese6. Each vcf was separated by chromosome (I and II) using bcftools (https://samtools.github.io/bcftools/bcftools.html). We then removed sites covered by a maximum of twice the mean chromosomal coverage using vcftools (https://vcftools.sourceforge.net/). For the Mentese6 alignment to *B. suis*, this was 6 or 5 reads for chromosome I and II, due to their differing mean coverage (2.62X and 2.26X respectively). We then masked any sites called as heterozygous in Mentese6 using bcftools.

To detect recombination in the modern data, we used MultiVCFAnalyzer v0.85.2[70] to produce a full genome alignment (including non-variable sites) of the modern genomes analysed (*n* = 226) and the *B. suis* reference for both chromosomes (Supplementary Data 5). The following parameters were used: a minimal genotyping quality of 30, a minimal coverage of 10 for base calls of homozygous alleles, and a minimal allele frequency of 80% for heterozygous calls. If any criteria was not met, an "N" was used in the respective genomic position. Recombination was detected in these modern data using Gubbins v3.1.0[71].

We then determined variant positions in *B. suis*-aligned modern and ancient genomes (including Mentese6; see Supplementary Data 5) with MultiVCFAnalyzer v0.85.2 with the following parameters: a minimal genotyping quality of 30, a minimal coverage of 3 for base calls, and a minimal allele frequency of 90% for heterozygous call. If any of the criteria were not met, an "N" was used in that respective genomic position. Recombinant regions detected by Gubbins in the previous step were masked using MultiVCFAnalyzer. We additionally computed coverage of Mentese6 in non-overlapping 50 bp sliding windows using bedtools coverage and masked regions in the SNP call with greater

than the 99th percentile coverage (windows with mean coverage > 10X). A total of 426 variant sites were masked by MultiVCFAnalyzer. Variant alignments produced by MultiVCFAnalyzer for chromosome I and II were concatenated using seqkit concat. We removed all positions in the resulting alignment where modern data contained an "N" (i.e. only sites where all modern samples had a base call were retained). The resulting alignment containing 30,364 varying sites and the reference genome was used to create a maximum-likelihood phylogeny with IQ-TREE2[67] using the nucleotide substitution model (TVM + F + ASC + R3) determined with modelfinder.

## Average nucleotide identity (ANI) using *B. suis* alignment

ANI calculation was performed using aniclustermap[27] on the full genome alignment of the *B. suis*-aligned sequences of *B. melitensis* and *B. abortus* after removing all sites with missing data ('N'). ANI values are reported in Supplementary Data 8, and Fig. 2 for values concerning Mentese6.

## Temporal signal

We tested for a temporal signal in our dataset of *B. suis*-aligned *B. melitensis* and *B. abortus* genomes using TempEst[28]. For the age of samples, the sampling year for modern genomes was used; for the 14th century mediaeval *B. melitensis*[16], the age of the death of the host was used; the Mentese6, the midpoint of its combined calibrated age was used. We found weak evidence for a temporal signal ($R^2 = 0.3837$, Supplementary Fig. 8). However, when sequences were assessed by their cladal identity within either species, there was a clear signal of local clock rates specific to subclades as expressed by clustering of clades by their root-to-tip distance. This rate variation appears to lead to a non-significant result in a dataset of limited pre-modern sequences (here, two). The Mentese6 sequence clearly shows a reduced root-to-tip distance (Fig. 2A and Supplementary Fig. 8), as expected by the radiocarbon-estimated age of the Mentese6 temporal bone.

## Beast time tree using *B. melitensis* alignment

To estimate the age of the most recent common ancestor of *B. melitensis* and *B. abortus* we produced a variant alignment of the Mentese6 sample, the *B. abortus* reference genome (GCF_000054005.1), and the 141 *B. melitensis* used in the "Phylogenetic analysis and reference datasets" section (Supplementary Data 5). We used the same pipeline as the used in the ML phylogeny step but instead aligned on *B. melitensis* (GCF_000007125.1), with the same filtering steps and alignment parameters as described for the *B. suis* alignment under "Phylogenetic analysis and reference datasets".

For genotyping, as coverage was higher for the *B. melitensis* alignment (mean coverage 3.62X and 3.17X for chromosomes I and II), we retained sites where Mentese6 had at least 3 reads and a maximum of 7 or 6 reads for chromosomes I and II respectively. We masked recombinant regions detected in modern genomes by Gubbins, regions with ≥ 99th percentile coverage as calculated (bedtools coverage) in non-overlapping 50 bp sliding windows (mean coverage > 11X), and sites where any modern genomes lacked a base call (i.e. "N" was allowed only in Mentese6). A total of 389 variants were masked by MultiVCFAnalyzer, producing an alignment with 18,300 varying sites.

BEAST v2.6.7[29] was used for divergence dating of the *B. melitensis*-*B. abortus* phylogeny. For modern genomes, collection dates were used as tip dates with a uniform distribution of 0.1 year around the collection date. One *B. melitensis* genome in the dataset (GCA_030370715.1) was obtained from a mediaeval Franciscan friar who died in 1394[16]. His death date was used as a tip date with a uniform distribution of 0.1 year around his death date. A lognormal distribution was used for the Mentese6 ancient sample tip date. We used the $C_{14}$ calibration (6057−5913 cal BCE/8007−7863 cal BP) as the M parameter and we varied the S parameter so the 95%/2σ lognormal range matched

the range in real space. The number of invariant sites was directly supplied in the xml file.

Beast model test[72] was used to test two different clock models, a lognormal relaxed clock and a strict clock, and two demographics models, coalescent exponential population and coalescent Bayesian skyline. We estimated marginal likelihoods to determine the best clock and demographic model for our dataset using path sampling. We estimated that a strict clock and coalescent Bayesian skyline demographic model were the best fit for our datasets by comparing each clock model and demographic model against each other (Supplementary Data 9). We then ran 3 replicates of 250 millions of chains using the mutation rate from a recent study[16] as the rate prior and with 1% burn-in. The BEAST parameters log file were assessed in Tracer v1.7.2 using 20% burn-in. This showed excellent mixing (ESS »300) and model convergence between runs. The tree files produced by BEAST were combined with logcombiner using 20% burn-in. Then we used TreeAnnotator, to produce a maximum clade credibility tree with 0% burn-in (20% burn-in was already applied by logcombiner).

## Variant effect annotation

*B. melitensis*-defining variants covered in Mentese6 and identified above ("Lineage analysis") were annotated using Ensembl Variant Effect Predictor (https://www.ensembl.org/info/docs/tools/vep/index.html) using the *B. melitensis* bv. 1 str. 16 M strain. Mentese6 genotype calls were first converted to an appropriate format and then uploaded to the Ensembl Variant Effect Predictor server (Supplementary Data 10). As variants were relative to this reference genome, we filtered for "missense_variant" and "stop_lost", the latter identifying nonsense mutations which had occurred in the *B. melitensis* lineage relative to other *Brucella*.

## Reporting summary

Further information on research design is available in the Nature Portfolio Reporting Summary linked to this article.

## Data availability

Collapsed sequencing data with host reads removed and *B. melitensis*-aligned bam file for Mentese prior to data QC are available at ENA accession PRJEB75678. Reference sequences used in this study are previously published and available for B. melitensis [https://www.ncbi.nlm.nih.gov/datasets/genome/GCF_000007125.1/], B. abortus [https://www.ncbi.nlm.nih.gov/datasets/genome/GCF_000054005.1/], and B. suis [https://www.ncbi.nlm.nih.gov/datasets/genome/GCF_000007505.1/]. We employed sequences previously analyzed by Abdel-Glil and colleagues [https://doi.org/10.1128/jcm.00311]; ENA accessions for each are presented in Supplementary Data 6. Source data are provided with this paper. The screened Menteşe specimens are under the temporary stewardship of Kevin G. Daly (UCD, Dublin) and permanent stewardship of Lionel Gourichon, Université Côte d'Azur, Nice. Applications to access and re-examine the material should be made to Dr. Gourichon. Specimen identifiers are provided in Data S1.

## Code availability

Custom code and scripts used in this manuscript are available on github [https://github.com/LouisLhote/Neolithic_Brucella_paper/] and via zenodo[73].

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

## Acknowledgements

We thank Daniel Bradley, Marco Rosario Capodiferro, Lara Cassidy, Evangelos Dimopoulos, Louis Du Plessis, Iseult Jackson, and Anahit Hovhannisyan for their advice on analysis and the manuscript. We thank Songül Alpaslan-Roodenberg, Daniel Helmer, and Jacob Roodenberg for their contributions as co-directors of excavations at Menteşe Höyük and for co-analysis of the faunal assemblage. We thank Andrew Hare for his contribution to laboratory work. This publication has emanated from research conducted with the financial support of Science Foundation Ireland under Grant number 21/PATH-S/9515(T) (to K.G.D.). M.D.T. was supported by European Research Council (ERC) Investigator Grant 295729-CodeX. Á.H. and V.M. are supported by the European Research Council under the European Union's Horizon 2020 research and innovation programme (885729-AncestralWeave). F.M.K. is supported by the Klaus Tschira Foundation (GSO/KT030) and together with I.L. supported by the Max Planck Society.

## Author contributions

K.G.D. conceived and supervised the study; L.L., V.M., M.D.T. and A.H. performed laboratory work; L.L. curated generated sequencing data and reference datasets; I.L. and F.M.K. conceived and developed the MALT database; L.L. processed and analysed the data with contributions from K.G.D. and F.M.K.; L.L. and K.G.D. wrote custom scripts and visualised data; L.G. provided the specimens; K.G.D. and L.L. wrote the initial draft with review and editing contributions from F.M.K., I.L., L.G, and all other co-authors.

## Competing interests

The authors declare no competing interests.
