## [Peer Review File · Nature Communications]

An 8,000 years old genome reveals the Neolithic origin of the zoonosis *Brucella melitensis*REVIEWER COMMENTS

Reviewer #1 (Remarks to the Author):

This is an interesting well-written paper that provides an exciting new perspective on the historical dynamics of zoonotic diseases. The work combines appropriate and sophisticated aDNA metagenomic and genomic analysis with archaeological context to uncover the deep history of *B. melitensis*. The authors report discovery of a 3.45X *B. melitensis* genome in an 8,000-year-old sheep specimen from Menteşe Höyük, Northwest Türkiye. They provide evidence for the genome's basal position relative to modern *B. melitensis*, and discuss its implications for understanding the pathogen's speciation and evolutionary timeline and its relevance to Neolithic livestock economies. The integration of coverage and breadth analyses, as well as the alignment of reads, demonstrates the robustness of the findings. The bioinformatic analyses to authenticate the ancient DNA appear robust. All I have to comment on is just a handful of clumsy phrases or typos.

Line 35: "with four species considered to be human-infecting²." should probably better said as "with four species considered to infect humans²."

Line 63:

"As infected females represent the main transmission source⁶, selective culling of young male goats evident in early herding communities could have created conditions permissive for sustained *Brucella* enzooticity¹¹," rewrite to make clear it is the culling that is evident.

Line 256:

"Despite *B. melitensis*' reduced capacity for respiration within the cell, it along with other core *Brucella* are traditionally considered to be facultative intracellular parasites." should be: "Despite *B. melitensis*' reduced capacity for respiration within the cell, it, along with other core *Brucella* species, is traditionally considered to be a facultative intracellular parasite."

Reviewer #2 (Remarks to the Author):

This study was superb, well written, and a pleasure to read. The authors present convincing data that the Menteşe⁶ sheep contains ancient *Brucella melitensis* DNA from the Neolithic period. Moreover, the study really helps nail down the timing of the emergence of the two most agriculturally and societally important *Brucella* species with the best supported data I have seen. Really exciting work.

Major Comments

None

Minor Comments (responses to most of these comments are not necessary)

L17. Italicize *melitensis*

L24. Speciation time from *B. abortus*? If so, be clear as this is important.

L26. I caution the use of word basal. Not that you are using it incorrectly but so many readers will infer basal as meaning older, when that is not necessarily true in many cases.

L36. Rephrase. It is not certain that *B. melitensis* is the most virulent to humans, *B. suis* quite likely is more virulent. Highest prevalence for sure.

L44. Suggest husbandry rather than keeping.

L51. The meat itself is an unlikely source of infection. Other tissues and organs such as lymph nodes, spleen, and liver for sure, but not the meat.

L59. Hypothesised by whom?

L69. The Jones paper is questionable. If brucellosis, it was almost certainly not *B. melitensis*. There's too much uncertainty in the conclusions/plausibility.

L152-154. Well phrased. But I do wonder if a conservative approach to keeping SNPs reduces diversity, making the branch shorter than it would actually be? Your data make sense, just

checking some of your assumptions.

L179. Lower ANI or higher ANI with W. Mediterranean clade? Or lower ANI because of the higher mutation rate? [Oh, I see. Disregard, you explain it more later in the text]. From Turkiye, one would expect E. Mediterranean lineage although the age of this sample likely predates any differentiation into the 3-4 main clades.

L206. This much more recent date of 4,317 BP is indeed surprising but your explanations are sound and plausible. Roughly 4000 years, almost half its "lifetime" without differentiating.

L216. Don't italicize reference 16

Supplemental figures

Supplementary Fig. 1. The legend indicates Mentese6 but the image clearly says Mentese 5. Why the discrepancy?

Supplementary Fig. 5. The percentages for the top 10 Reference Genomes assignments are awkwardly written, e.g. 099%

Supplementary Fig. 7. Nice figure.

REVIEWER COMMENTS

Reviewer #1 (Remarks to the Author):

This is an interesting well-written paper that provides an exciting new perspective on the historical dynamics of zoonotic diseases. The work combines appropriate and sophisticated aDNA metagenomic and genomic analysis with archaeological context to uncover the deep history of *B. melitensis*. The authors report discovery of a 3.45X *B. melitensis* genome in an 8,000-year-old sheep specimen from Menteşe Höyük, Northwest Türkiye. They provide evidence for the genome's basal position relative to modern *B. melitensis*, and discuss implications for understanding the pathogen's speciation and evolutionary timeline and its relevance to Neolithic livestock economies. The integration of coverage and breadth analyses, as well as the alignment of reads, demonstrates the robustness of the findings. The bioinformatic analyses to authenticate the ancient DNA appear robust. All I have to comment on is just a handful of clumsy phrases or typos.

We thank the reviewer for their time and effort taken for this review, as well as the positive appraisal and constructive comments.

In the interests of clarity we note that the methods section missed a tree-thinning step, as indicated to the editor prior to peer review. Explanatory text has now been added (lines 499-502 in manuscript docx with figures) reading "and then removed sequences which lacked a known sampling year and after thinning the dataset with treemmer (-RTL 0.95; available at <https://github.com/fmenardo/Treemmer>) following the approach of Long and colleagues^{15,16}".

Line 35: "with four species considered to be human-infecting²." should probably better said as "with four species considered to infect humans²."

We have made the recommended change (lines 36-37).

Line 63:

"As infected females represent the main transmission source⁶, selective culling of young male goats evident in early herding communities could have created conditions permissive for sustained *Brucella* enzooticity¹¹," rewrite to make clear it is the culling that is evident.

This line now reads "selective culling of young male goats practised in early herding communities" (line 65).

Line 256:

"Despite *B. melitensis*' reduced capacity for respiration within the cell, it along with other core *Brucella* are traditionally considered to be facultative intracellular parasites." should be: "Despite *B. melitensis*' reduced capacity for respiration within the cell, it, along with other core *Brucella* species, is traditionally considered to be a facultative intracellular parasite."

We have made the recommended change (lines 254-256).

Reviewer #2 (Remarks to the Author):

This study was superb, well written, and a pleasure to read. The authors present convincing data that the Menteşe sheep contains ancient *Brucella melitensis* DNA from the Neolithic

period. Moreover, the study really helps nail down the timing of the emergence of the two most agriculturally and societally important *Brucella* species with the best supported data I have seen. Really exciting work.

We also thank the reviewer for their time and effort taken for this review, and their thoughtful comments.

In the interests of clarity we note that the methods section missed a tree-thinning step, as indicated to the editor prior to peer review. Explanatory text has now been added (lines 499-502 in manuscript docx with figures) reading “and then removed sequences which lacked a known sampling year and after thinning the dataset with treemmer (-RTL 0.95; available at <https://github.com/fmenardo/Treemmer>) following the approach of Long and colleagues^{15,16}” .

Major Comments

None

Minor Comments (responses to most of these comments are not necessary)

L17. Italicize melitensis

We have made the recommended change.

L24. Speciation time from *B. abortus*? If so, be clear as this is important.

Yes, we were referring to the speciation time from *B. abortus*. To be more precise the abstract now reads “allows the calibration of the *B. melitensis* speciation time from the primarily cattle-infecting *B. abortus*” (lines 24-25) .

L26. I caution the use of word basal. Not that you are using it incorrectly but so many readers will infer basal as meaning older, when that is not necessarily true in many cases.

We appreciate the reviewers concern, as we intended to be as precise as possible in our use of the term “basal”. In this instance we agree it is appropriate given the phylogenetic results and temporal context. To improve the clarity we have edited the text to read “genome is basal with respect to all known *B. melitensis*” (line 23).

L36. Rephrase. It is not certain that *B. melitensis* is the most virulent to humans, *B. suis* quite likely is more virulent. Highest prevalence for sure.

We have changed the line to be more precise regarding the prevalence of different *Brucella* species. This line now reads “Of these, *Brucella melitensis* is the most frequent cause of brucellosis in” (lines 37-38).

L44. Suggest husbandry rather than keeping.

We have amend the word “keeping” to “husbandry” (line 45)

L51. The meat itself is an unlikely source of infection. Other tissues and organs such as lymph nodes, spleen, and liver for sure, but not the meat.

We have removed the mention of meat. The line now reads “as is the unpasteurised milk of infected animals” (line 52).

L59. Hypothesised by whom?

The association with the beginning of livestock herding and the epidemiological transition or evolution of *Brucella* has been referred to obliquely in several papers (D’Anastasio et al

2010, DOI: 10.1017/S095026881000097X, Moreno 2014, DOI: 10.3389/fmicb.2014.00213), and was most explicitly linked in Fournié et al 2017 (DOI: 10.1098/rsos.160943). As such we have added the Fournié reference to this line.

L69. The Jones paper is questionable. If brucellosis, it was almost certainly not *B. melitensis*. There's too much uncertainty in the conclusions/plausibility.

The review makes a reasonable point on this citation. We have replaced it with an alternative reference documenting palaeopathological and palaeogenetic evidence human brucellosis (Mutolo et al, 2012, DOI: 10.1002/ajpa.21643).

L152-154. Well phrased. But I do wonder if a conservative approach to keeping SNPs reduces diversity, making the branch shorter than it would actually be? Your data make sense, just checking some of your assumptions.

As the review will appreciate, filtering criteria creates a trade-off between true/false positive variant rates, together with the false negative rate: the more conservative we are, the more likely we remove spurious variant sites while potentially discarding true variant sites. For this particular study, alignment from soil-living *Ochrobactrum* could artificially increase the divergence observed for the Mentese6 *Brucella*, affecting the divergence time estimates. The high coverage at certain gene regions (Figure 1C) indicate spurious as a potential source of error (hence our application of a coverage-window site filter). We also observed very low rates of *B. inopinata* allele matching in our lineage analysis (Figure S7). As such we erred in this trade off to minimise these false positive risks, a typical concern in ancient pathogen studies but of particular importance in the context of the genomically homogenous *Brucella* and a modest genome coverage.

L179. Lower ANI or higher ANI with W. Mediterranean clade? Or lower ANI because of the higher mutation rate? [Oh, I see. Disregard, you explain it more later in the text]. From Turkiye, one would expect E. Mediterranean lineage although the age of this sample likely predates any differentiation into the 3-4 main clades.

Disregarded. As the reviewer notes, it would appear that it is the W. Mediterranean clade shows a higher mutation rate relative to other lineages, which drives this pattern, rather than phylogeographic structure.

L206. This much more recent date of 4,317 BP is indeed surprising but your explanations are sound and plausible. Roughly 4000 years, almost half its "lifetime" without differentiating. We appreciate the reviewer's assessment of this result and our interpretation. The nature of *Brucella* as a clonal intracellular species necessitates engagement with the demographic history of the host. While associated with goat today, it may be that over millennia time scales it was instead sheep which drove the apparently-recent MRCA of *B. melitensis*.

L216. Don't italicize reference 16

We have made the recommended change.

Supplemental figures

Supplementary Fig. 1. The legend indicates Mentese6 but the image clearly says Mentese 5. Why the discrepancy?

There was a duplication of labelling efforts within the institution this work was completed at. Five of the Mentese samples were initially labelled 1-5, and then redesignated Mentese6-11 (see Table S1). The underlying archaeological code (MS1) is maintained between these labelling systems. To clarify, we have added to the Supplementary Figure 1 legend “Mentese5” refers to a deprecated labelling system.”.

Supplementary Fig. 5. The percentages for the top 10 Reference Genomes assignments are awkwardly written, e.g. 099%

We agree with the reviewer that it is not ideal; it is the standard output for the HOPS. This formatting is seen on the example output on the HOPS github (https://github.com/keyfm/amps/blob/master/profilePDF_explained.pdf). We opted to present the unaltered HOPS output somewhere in materials of the manuscript rather than replot all data (such as in Figure 1B). We have added additional text to the figure caption: “The percentage values in the table contain a superfluous zero i.e. 099% = 99%”.

Supplementary Fig. 7. Nice figure.

We thank the reviewer for their comment.